# Inversion symmetry and bulk Rashba effect in methylammonium lead iodide perovskite single crystals

Kyle Frohna [1,2], Tejas Deshpande [3], John Harter[3], Wei Peng[4], Bradford A. Barker[5,6], Jeffrey B. Neaton[5,6,7], Steven G. Louie[5,6], Osman M. Bakr[4], David Hsieh[3] & Marco Bernardi[1]

Methylammonium lead iodide perovskite ($MAPbI_3$) exhibits long charge carrier lifetimes that are linked to its high efficiency in solar cells. Yet, the mechanisms governing these unusual carrier dynamics are not completely understood. A leading hypothesis—disproved in this work—is that a large, static bulk Rashba effect slows down carrier recombination. Here, using second harmonic generation rotational anisotropy measurements on $MAPbI_3$ crystals, we demonstrate that the bulk structure of tetragonal $MAPbI_3$ is centrosymmetric with *I4/mcm* space group. Our calculations show that a significant Rashba splitting in the bandstructure requires a non-centrosymmetric lead iodide framework, and that incorrect structural relaxations are responsible for the previously predicted large Rashba effect. The small Rashba splitting allows us to compute effective masses in excellent agreement with experiment. Our findings rule out the presence of a large static Rashba effect in bulk $MAPbI_3$, and our measurements find no evidence of dynamic Rashba effects.

[1] Department of Applied Physics and Materials Science, California Institute of Technology, Pasadena, CA 91125, USA. [2] School of Physics, Trinity College Dublin, Dublin 2, Ireland. [3] Department of Physics, California Institute of Technology, Pasadena, CA 91125, USA. [4] Division of Physical Sciences and Engineering, KAUST Catalysis Center (KCC), King Abdullah University of Science and Technology (KAUST), Thuwal 23955-6900, Saudi Arabia. [5] Department of Physics, University of California, Berkeley, CA 94720-7300, USA. [6] Materials Science Division, Lawrence Berkeley National Laboratory, Berkeley, CA 94720, USA. [7] Molecular Foundry, Lawrence Berkeley National Laboratory, Berkeley, CA 94720, USA. These authors contributed equally: Kyle Frohna, Tejas Deshpande, John Harter. Correspondence and requests for materials should be addressed to D.H. (email: dhsieh@caltech.edu) or to M.B. (email: bmarco@caltech.edu)

ntense research efforts have enabled a dramatic increase in the power conversion efficiency of methylammonium lead iodide ($MAPbI_3$)[1] solar cells[2–5]. These rapid advances relied on the many desirable properties of $MAPbI_3$, including its easy deposition from solution[2,6] or vapor[7], a near-ideal band gap[8,9] that can be tuned with composition[10,11], and strong sunlight absorption in sub-micrometre thick films[12]. Despite the rise in device performance, the origin of the exceptional transport properties of $MAPbI_3$ remains the subject of debate. Long carrier diffusion lengths of up to 1 μm have been recognized early on as a key factor for device performance[13], and diffusion lengths as high as tens of microns in single crystals (and more recently in polycrystalline thin films)[14–16] have been reported. The associated long carrier lifetimes of up to tens of microseconds are also particularly surprising given that $MAPbI_3$ films are solution processed and polycrystalline[14,17].

Understanding carrier dynamics in $MAPbI_3$ requires a close examination of its atomic and electronic structure, which are unusually complex compared to conventional solar cell materials[18]. Similar to other perovskites, crystals of $MAPbI_3$ exhibit multiple phases: cubic with a space group of $Pm\bar{3}m$ above 327 K[19], tetragonal with an elongated $c$ axis and alternating in-plane rotations of the $PbI_6$ octahedra between 162 and 327 K[19], and orthorhombic with a space group of $Pnma$ below 162 K[20]. The rotations of the methylammonium (MA) ions[19,21] and their interactions with the lead iodide (PbI) framework further complicate the structure. The space group of the room temperature tetragonal phase—the most relevant for practical applications—has been the subject of extensive debate, with some works reporting a centrosymmetric space group of $I4/mcm$ (point group $4/mmm$)[19,20] and others a non-centrosymmetric space group of $I4cm$ (point group $4mm$)[22,23].

The presence of heavy elements such as lead and iodine introduces a strong spin–orbit coupling (SOC) in the electronic structure. When inversion symmetry is absent, as is the case in the bulk of non-centrosymmetric crystals and at surfaces or interfaces, the electrons feel an effective magnetic field due to the SOC. This interaction, known as the Rashba effect[24,25], removes the electron spin degeneracy and splits each of the valence and conduction band edges, resulting in an indirect band gap. Recent work proposed that a bulk Rashba effect—i.e., not due to the surface or interfaces—may be the cause of long carrier lifetimes and diffusion lengths in $MAPbI_3$[26–28]. This idea has received surprising attention in spite of the fact that, with the exception of a few reports on the tetragonal phase[22,23], most diffraction experiments attribute centrosymmetric space groups to all three crystal phases[19,20].

There are two main mechanisms that can break inversion symmetry in the bulk of $MAPbI_3$, one static and one dynamic. The PbI framework can statically distort from a set of ideal corner-sharing octahedra, forming a non-centrosymmetric phase as has been recently suggested[23], and resulting in a static bulk Rashba effect. Alternatively, inversion symmetry can be broken dynamically by the MA ion rotations in the absence of a static distortion of the PbI framework. Since the point group of the MA ions is $C_{3v}$, the relatively free MA ion rotation at room temperature[19] implies that the highly symmetric point groups assigned to $MAPbI_3$ may actually be approximations of a more complex dynamic structure[19,20]. Dynamic structural fluctuations can also occur due to phonon modes or the interaction of the MA ions with the PbI framework[28–30]. All these scenarios can induce a so-called dynamic bulk Rashba effect with a characteristic time scale shorter than those of the MA ion rotation (~10 ps)[21] or the vibration of the highest-frequency PbI framework optical mode (~250 fs)[31,32]. The strong interplay between the SOC, the PbI framework and the MA ion rotation intertwines the structural, electronic and vibrational degrees of freedom in $MAPbI_3$.

In this work, we set out to resolve the debate on the tetragonal space group and the related bulk Rashba effects by combining second harmonic generation rotational anisotropy (SHG-RA) experiments, which have recently been established as a sensitive probe of point group symmetry[33], and density functional theory (DFT) calculations. We find that the SHG-RA data fits excellently to the centrosymmetric $I4/mcm$ space group, and perform several sets of DFT calculations that establish the absence of a large bulk Rashba effect in tetragonal $MAPbI_3$. When the PbI framework is constrained to the observed $I4/mcm$ symmetry and only the MA ions are relaxed, we find a very small Rashba splitting due to the random MA ion orientation, making the band gap nearly direct and the band edge dispersions nearly parabolic. However, when two widely used approaches to relax tetragonal and cubic cells in DFT are employed, non-centrosymmetric crystal structures with distorted PbI framework and atomic positions inconsistent with experiment are formed. The large Rashba effect found in these structures is shown to be an artifact of DFT. The SHG-RA measurements do not find evidence of a dynamic Rashba effect, and the computational results show that the MA ion rotations can generate only a small dynamic Rashba effect. Our results unravel complex structure–property relationships in tetragonal $MAPbI_3$, providing evidence of its centrosymmetric nature, direct band gap, and negligible bulk Rashba effect.

## Results

**Experimental determination of the point group of $MAPbI_3$.** While it is difficult to distinguish between the centrosymmetric $I4/mcm$ and non-centrosymmetric $I4cm$ space groups with diffraction based techniques, the presence or absence of inversion symmetry has dramatic consequences on SHG. The bulk electric dipole (ED)-induced SHG is only allowed in non-centrosymmetric crystals[34]; in centrosymmetric crystals, the ED-induced SHG is forbidden in the bulk due to inversion symmetry, but still allowed at the surface where inversion symmetry is necessarily broken. Therefore it is necessary to determine whether an SHG signal has surface or bulk origin using a technique like SHG-RA, which was not done in ref. [23], before drawing conclusions about whether or not the bulk crystal structure breaks inversion symmetry. We perform SHG-RA measurements on single crystals of $MAPbI_3$ in the tetragonal phase at room temperature (see Methods) to determine its point group symmetry, and compare the SHG-RA data with both the bulk ED response expected from a $I4cm$ crystal and the surface ED response from a $I4/mcm$ crystal, which has surface point group $mm2$. The space groups and associated point groups are summarized in Table 1.

Figure 1 shows the SHG-RA results for different combinations of incident (in) and outgoing (out) polarizations, both of which can be either parallel (P) or perpendicular (S) to the light scattering plane, and compares the experimental data to simulated SHG-RA patterns calculated based on the structure

**Table 1 Space groups and corresponding point groups for tetragonal $MAPbI_3$**

| Bulk space group | Point group | Electric dipole SHG allowed? |
|---|---|---|
| $I4/mcm$ (centrosymmetric) | $4/mmm$ (bulk) | No |
| | $mm2$ (surface) | Yes |
| $I4cm$ (non-centrosymmetric) | $4mm$ (bulk) | Yes |

Proposed tetragonal $MAPbI_3$ space groups, their corresponding surface and bulk point groups and whether the point group allows ED SHG

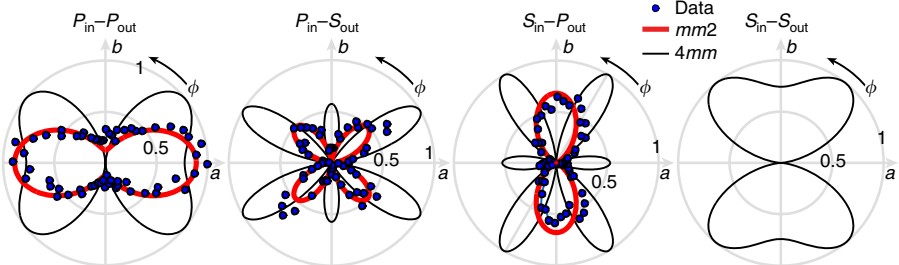

**Fig. 1** SHG rotational anisotropy experiments. Polar plots of the SHG-RA data (blue dots) from the (100) surface of MAPbI$_3$, where each panel shows results for a different set of incident and outgoing polarizations. The radial direction represents the normalized magnitude of the SHG intensity and the azimuthal angle $\phi$ the orientation of the light scattering plane relative to the tetragonal $a$-axis. The expected electric-dipole SHG radiation patterns from the bulk of a *I4cm* crystal (black) and best fits of the data to electric-dipole SHG radiation patterns from the surface of a *I4/mcm* crystal (red) are superposed on top of the data. In the latter case, SHG is forbidden by symmetry in $S_{in}-S_{out}$ geometry

of the ED induced SHG susceptibility tensor for the *mm*2 and 4*mm* point groups[33,35].

For all polarization channels, the data fit excellently the angular dependence of the SHG intensity for the surface *mm*2 point group of a crystal with space group *I4/mcm*. On the other hand, the expected response from a crystal with space group *I4cm* is qualitatively incompatible with the measurements. In particular, the measured number and orientation of the lobes in the $P_{in}-P_{out}$, $P_{in}-S_{out}$, and $S_{in}-P_{out}$ channels and the absence of any SHG intensity in the $S_{in}-S_{out}$ channel cannot be reconciled with a 4*mm* point group. This result is consistent with several diffraction experiments[19,20], and directly disproves the recently hypothesized non-centrosymmetric space group *I4cm*[22,23]. On this basis, given the centrosymmetric nature of MAPbI$_3$, the static bulk Rashba effect is expected to be negligible.

The resolution of our SHG-RA measurements allows us to additionally comment on dynamic Rashba effects. If there were dynamic inversion symmetry-breaking due to the MA ions[36] or the PbI framework phonons[31,32], then one would expect the SHG contribution from a 4*mm* point group to fluctuate between zero and some finite value, resulting in a non-zero time-averaged signal. Given that a complete SHG-RA scan takes several minutes, some admixture of the *mm*2 and 4*mm* contributions shown in Fig. 1 would be expected, with the latter likely being dominant since bulk ED signals are usually far stronger than surface signals. Given that we do not detect any 4*mm* contribution above our noise floor, we find no evidence of dynamic inversion symmetry breaking. With regards to the possibility of spatial domains, the SHG-RA patterns shown in Fig. 1 were reproducible upon changing the beam position over the entire surface of the sample. Therefore we find no evidence of 4*mm* domains on the length scale of our resolution of ~10 microns.

**Ab initio structural relaxation of MAPbI$_3$.** The presence of bulk inversion symmetry in tetragonal MAPbI$_3$ is crucial to correctly predict its properties using first-principles calculations. The computational determination of crystal structures is a difficult problem that is rich with subtleties. Starting from a guessed initial structure, an ideal computational approach would be able to relax the atomic positions of tetragonal MAPbI$_3$ to our measured space group symmetry. DFT calculations have been used extensively to study the structure of MAPbI$_3$. However, since DFT is inherently a ground state (zero-temperature) method, it cannot be applied straightforwardly to obtain the structure of high-temperature phases (such as the tetragonal and cubic phases of MAPbI$_3$), which are stabilized by vibrational entropy. For example, the so-called vibrational soft modes in perovskite materials[37], which possess frequencies that vary continuously with temperature and

exhibit peculiar trends at phase transitions, cannot be accurately described in DFT phonon calculations. This fact is a clear indication that probing the potential energy surface at zero temperature, as is done in DFT structural relaxations, is inadequate to compute the structures of the tetragonal and cubic MAPbI$_3$ phases. While ab initio molecular dynamics based on DFT can capture thermal effects, the accurate first-principles methods available to compute bandstructures, absorption spectra and more recently carrier transport are applied on a snapshot of the structure, which clearly needs to be representative of the space group symmetry of the material. Despite these complexities and pitfalls, DFT structural relaxations of tetragonal and cubic MAPbI$_3$ have been used extensively as a starting point to compute their bandstructures[38–41].

To study the impact of the structure on bandstructure and Rashba effect in tetragonal and cubic MAPbI$_3$, and understand the importance of using structure snapshots with the correct space group symmetry measured here, we carry out DFT structural relaxations using a variety of protocols, and then compute the bandstructure and bulk Rashba effect. We first relax initial tetragonal and cubic MAPbI$_3$ structures without constraining the symmetry and lattice parameters during the relaxation, as is commonplace in the literature[38–41]. These calculations employ $\sqrt{2}\times\sqrt{2}\times 2$ cells with four formula units; each of the four MA ions is randomly aligned in one of the equivalent $\langle 111 \rangle$ directions[19], and the local-density approximation (LDA) of DFT is employed (see Methods).

Figure 2a, b shows the lattice parameters as a function of relaxation step for such cubic and tetragonal structures, respectively. In both phases, the three lattice parameters initially shrink by 1–2% compared to the initial (experimental) value, a well-known trend for the LDA. As the relaxation continues, the three lattice parameters become distinct and both the initially tetragonal and cubic structures become orthorhombic, with three different lattice constants and nearly orthogonal lattice vectors. We interpret this result as a consequence of the lack of vibrational entropy in the calculation, and as an attempt by DFT to capture the zero-temperature orthorhombic phase of MAPbI$_3$. In these fully relaxed structures (see Fig. 2c, d) the Pb atoms move from the center of the octahedra, distorting the PbI framework and resulting in alternating long and short Pb−I bonds. This off-centering of the Pb atoms breaks inversion symmetry and generates a large structural distortion. While in previous work such orthorhombic structures have been described as pseudo-cubic[26,41] and employed to investigate the electronic structure of tetragonal and cubic MAPbI$_3$, it is clear that the DFT ground state is entirely different from that of these high-temperature phases, which are stabilized by vibrational entropy. In a second set of DFT relaxations, the lattice parameters are fixed but the

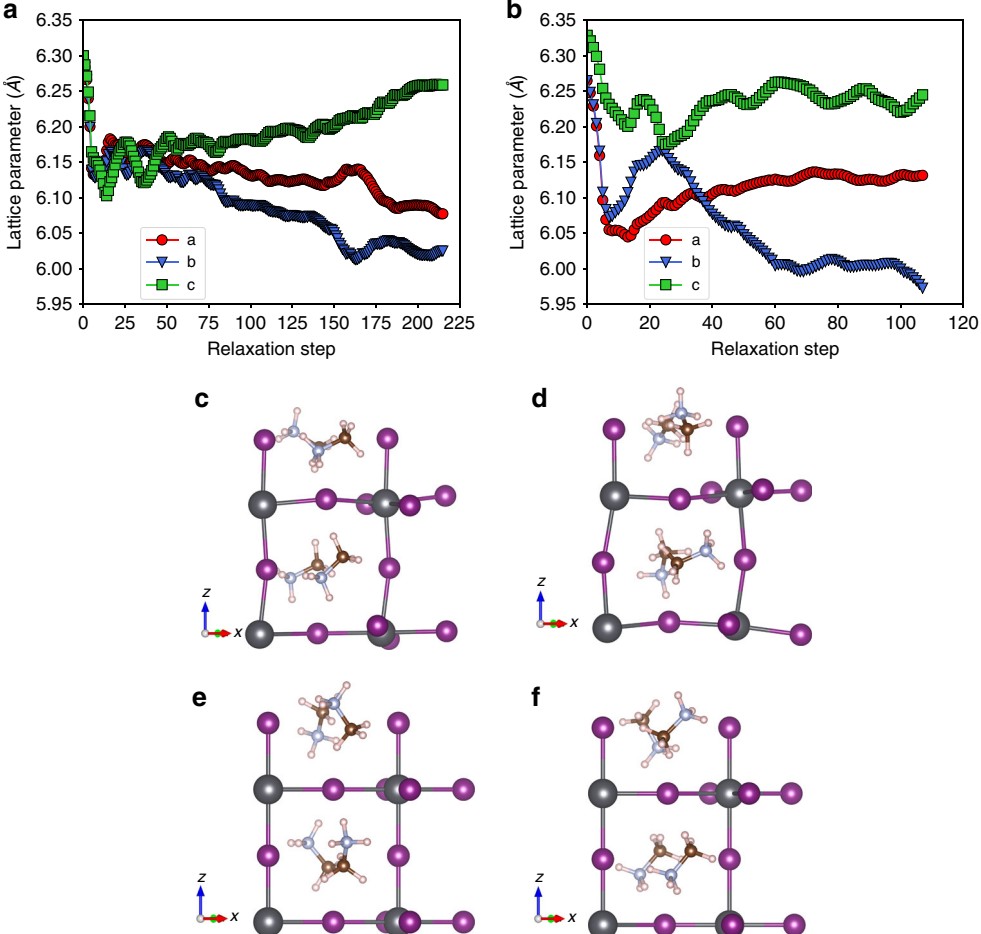

**Fig. 2** Structural relaxation in DFT. Lattice parameters (labelled **a**, **b**, **c**) as a function of relaxation step for cubic (**a**) and tetragonal (**b**) MAPbI$_3$ structures. The out-of-plane lattice parameter is denoted as $c$ in the tetragonal structure. The relaxation results in both structures becoming orthorhombic, as seen by the different values of the three lattice parameters at the end of the relaxation. The cubic (**c**) and tetragonal (**d**) structures obtained after full relaxation without symmetry constraints exhibit considerable distortions of the PbI framework. The MA-relaxed cubic (**e**) and tetragonal (**f**) structures with symmetry consistent with experiment are also shown. All structures consist of 4-formula-unit supercells, and the atoms are represented with spheres: Pb is grey, I is purple, C is brown, N is blue, and H is pink

atoms are allowed to relax, as is also commonplace in the literature[26,27,39]. The resulting structures (see Supplementary Fig. 1) are similar to the fully relaxed case, combining Pb off-centering, a distorted PbI framework, and extensive inversion symmetry breaking. Note that if the low temperature orthorhombic structure is fully relaxed, the octahedra remain undistorted [42]. We verify this result by fully relaxing an experimentally measured orthorhombic structure[43], and find that the structure remains centrosymmetric upon relaxation (see Supplementary Data 3 and the related bandstructure in Supplementary Fig. 2).

**Bandstructure of cubic and tetragonal MAPbI$_3$.** Next, we adopt an alternative procedure to obtain high-temperature MAPbI$_3$ structures where the $I4/mcm$ tetragonal and $Pm\bar{3}m$ cubic symmetries consistent with experiment are enforced by fixing the PbI framework and allowing only the MA ions to relax (see Methods). The resulting structures, referred to in the following as MA-relaxed, are shown in Fig. 2e, f (data files of example MA-relaxed structures are provided in Supplementary Data 1 and 2). The electronic bandstructures of the fully relaxed and MA-relaxed structures are compared in Fig. 3 for both single-formula-unit cells and 4-formula-unit cells with each of the 4 MA ions aligned

randomly in one of the ⟨111⟩ equivalent directions (see Methods). Note that the PbI framework is non-centrosymmetric in the fully relaxed structures and centrosymmetric in the MA-relaxed structures, which are however overall non-centrosymmetric due to the random MA orientations. The bandstructures of the fully relaxed tetragonal and cubic structures exhibit a large Rashba splitting of the band edges. By contrast, in the MA-relaxed structures the splitting is considerably reduced in all cases, and the residual Rashba effect is due to the parallel alignment (in the single-formula-unit cell in Fig. 3a, b) and random orientation (in the supercell in Fig. 3c) of the MA ions in our simulation cells. We expect that in the real crystal the net electric field from many randomly oriented polar MA ions, and thus the bulk Rashba effect, would be on average even smaller than that in our calculations on cells with only a handful of MA ions.

Given that the LDA plus SOC significantly underestimates the band gap, we re-compute the bandstructure for the single-formula-unit MAPbI$_3$ cubic structure using the Heyd-Scuseria-Ernzerhof (HSE) hybrid functional (see Methods)[44], which can improve the accuracy of the computed bandstructure. Figure 3b shows that the qualitative behavior of the Rashba splitting in the fully relaxed and MA-relaxed structures is the same in the LDA and HSE calculations, a clear sign that the Rashba splitting is a consequence of the structural distortions of the PbI framework,

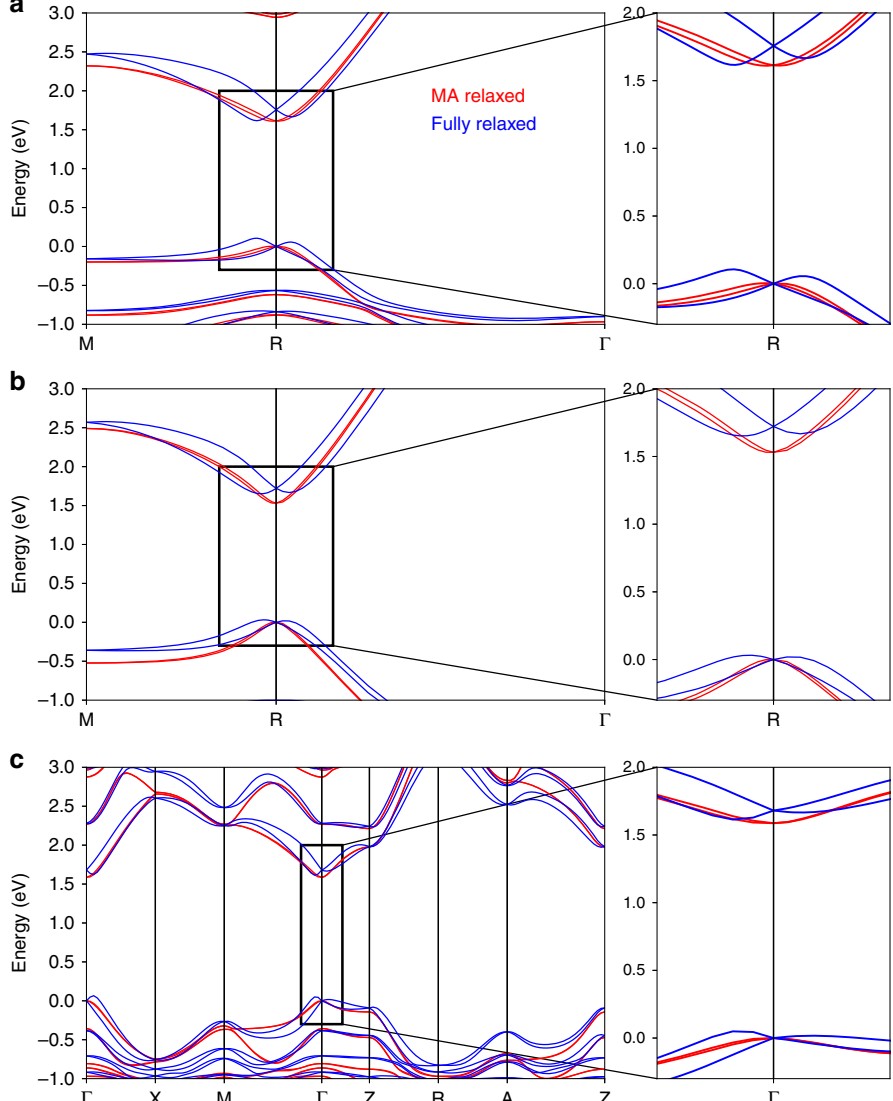

**Fig. 3** Structural effects on the bandstructure of MAPbI₃. Comparisons between the bandstructures of fully relaxed (blue) and MA-relaxed (red) MAPbI₃ structures. In **a**, **b**, the bandstructures are shown for single-formula-unit cubic structures with the MA ions aligned in the [100] direction, computed within the LDA (**a**) and HSE (**b**) approximations of DFT. **c** Bandstructures for a 4-formula-unit tetragonal supercell. The Rashba splitting, which is magnified in the right panel insets, is in all cases much greater in the fully relaxed structures due to the large PbI framework distortions that break inversion symmetry. The *k*-path labeling of the tetragonal cell is taken from the ref. [65]

regardless of the approximation employed to describe electron interactions. Employing centrosymmetric snapshots of the structure is clearly important since the magnitude of the Rashba splitting is fictitiously large in the fully relaxed structures with non-centrosymmetric PbI framework.

In the bandstructures in Fig. 3, the bulk Rashba effect removes the spin degeneracy of both the valence and conduction bands, leading to the formation of distinct parabolic band minima shifted from a high-symmetry point of the Brillouin zone (R point in the single-formula-unit cell and Γ point in the 4-formula-unit supercells). This trend can be captured by a Rashba Hamiltonian ($H_\mathrm{R}$) acting on each **k**-point with an effective SOC interaction[24]:

$$H_\mathrm{R} = \alpha(\mathbf{k} \times \boldsymbol{\sigma}) \cdot \hat{\mathbf{n}}, \qquad (1)$$

where $\boldsymbol{\sigma}$ are the Pauli spin matrices, and $\hat{\mathbf{n}}$ is a unit vector along the electric field induced by the breaking of inversion symmetry. The Rashba parameter $\alpha$, which measures the magnitude of the splitting, depends on the crystallographic direction and on the

magnitude of the macroscopic polarization due to inversion symmetry breaking[45]. The resulting low-energy bandstructure along a given direction consists of two split parabolic bands with energy $E_\pm(k) = \frac{\hbar^2 k^2}{2m^*} \pm \alpha k$ and, for the conduction band, with minima at $\mathbf{k}_m = \alpha \times m^*/\hbar^2$ (where $m^*$ is the effective mass), from which the Rashba parameter can be extracted using $\alpha = 2|E_-(k_m)|/k_m$. Analogous expressions hold for the valence band and its Rashba parameter. A system that is only slightly perturbed from its centrosymmetric structure—such as our MA-relaxed structures—exhibits a small value of $\alpha$, and accordingly a small band splitting[24].

**Quantifying the Rashba effect**. To quantify the bulk Rashba splitting in MAPbI₃, we first compute the bandstructures along high symmetry directions for 10 tetragonal and 10 cubic 4-formula-unit MA-relaxed supercells. In each structure, the 4 MA ions are initially randomized in one of the 8 equivalent ⟨111⟩ directions and then relaxed, thus allowing us to extract a

**Table 2 Computed Rashba parameters**

|  | CB (cubic) | VB (cubic) | CB (tetragonal) | VB (tetragonal) |
|---|---|---|---|---|
| $\langle 100 \rangle$ $(\Gamma X)$ | 0.16 ± 0.11 | 0.13 ± 0.11 | 0.45 ± 0.21 | 0.40 ± 0.23 |
| [110] $(\Gamma M)$ | 0.17 ± 0.14 | 0.18 ± 0.11 | 0.47 ± 0.21 | 0.40 ± 0.17 |
| [111] $(\Gamma R/\Gamma A)$ | 0.16 ± 0.12 | 0.15 ± 0.13 | 0.45 ± 0.23 | 0.39 ± 0.21 |
| [001] $(\Gamma Z)$ | — | — | 0.23 ± 0.19 | 0.17 ± 0.18 |

Rashba parameters ($\alpha$) in units of eV Å computed along different high-symmetry directions for both cubic and tetragonal MA-relaxed structures. The mean $\alpha$ values for 10 cubic and 10 tetragonal structures with random MA orientation are given, together with the associated standard deviations. The Rashba coefficient in the $\langle 100 \rangle$ direction is an average of the coefficients computed along the [100], [010] and [001] directions. The directions refer to a 4-formula-unit simulation cell with the band edges at the $\Gamma$ point of the Brillouin zone. CB is the conduction band, and VB the valence band

distribution of Rashba parameters and effective masses over multiple sets of MA ion orientations[19]. In each of these MA-relaxed structures, the PbI framework is fixed to the experimental symmetry and only the MA ions are allowed to relax. The average and standard deviation of the Rashba parameters for these structures are given in Table 2, and an example bandstructure is given in Fig. 3c.

Our computed Rashba parameters are of order (0.1–0.2) ± 0.1 eV Å in the cubic structures and (0.2–0.4) ± 0.2 eV Å in the tetragonal structures. These values are an order of magnitude smaller than those of fully relaxed structures with large distortions of the PbI framework. For example, ref. [41] reported $\alpha \approx 1$ eV Å in fully relaxed cubic structures, and ref. [25] a value of $\alpha = 4$ eV Å in a distorted cubic structure with $P4mm$ space group. While refs. [26,40] did not compute the Rashba coefficient of their fully relaxed structures, a large Rashba splitting can be seen in their bandstructures. We conclude that once the experimental crystal structure is maintained, the bulk Rashba effect is much smaller than previously predicted and only due to the random MA ion orientations, leading to a nearly direct band gap for tetragonal and cubic MAPbI$_3$ and to an expected negligible effect of SOC on charge transport. In fact, since carriers undergo diffusive transport with scattering events spaced apart by tens of unit cells, they feel a space and time averaged structure, and are insensitive to the small Rashba effect induced by the local MA ion orientations. Lastly, note that the bandstructure of orthorhombic MAPbI$_3$ similarly exhibits a negligible Rashba effect (see Supplementary Fig. 2).

## Discussion
The presence of a surface Rashba effect is clearly not ruled out by our work, given that inversion symmetry is inherently broken at surfaces and interfaces; its role in devices may be significant[46]. Since inversion symmetry is also broken at grain boundaries, Rashba effects in polycrystalline films may also differ greatly from our single crystal results. The influence of Rashba effects due to surfaces, interfaces and grain boundaries depends on the penetration depth of the surface electric field that couples to the electrons through the SOC. The Debye length ($L_D$) provides an estimate of the length scale over which such surface Rashba effects are significant:

$$L_D = \sqrt{\frac{\epsilon_0 \epsilon_\infty k_B T}{e^2 n}}, \qquad (2)$$

where $\epsilon_0$ and $\epsilon_\infty$ are the dielectric constant of the vacuum and of the material, respectively, $k_B$ the Boltzmann constant, $T$ is temperature, $e$ the electron charge and $n$ the carrier concentration. Using the reported room temperature high frequency dielectric constant of 23.3[47], and an intrinsic charge carrier density of $1 \times 10^9$ cm$^{-3}$[22], we find a Debye length value of ~180 μm in the dark, and ~180 nm under solar illumination assuming the carrier

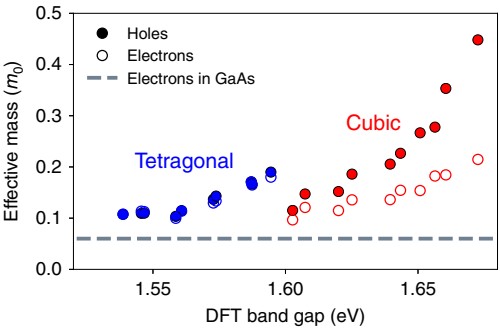

**Fig. 4** Carrier effective masses. Effective masses of electrons and holes in the cubic (red) and tetragonal (blue) MAPbI$_3$ phases versus the band gap, shown for 10 structures (in each phase) that differ only for their MA random alignment. The dotted line given for comparison is the electron effective mass of GaAs. The LDA band gaps were shifted by ~1.2 eV to match the experimental gap

concentration reaches ~$10^{15}$ cm$^{-3}$[48]. Surface Rashba effects could thus be particularly relevant in solar cells, where perovskite films are typically several hundred nanometers thick[49,50], and in polycrystalline thin films, where typical grain sizes in high-performing devices are of order 100 nm–1 μm[49,50]. Our results imply that the source of any Rashba effects that may have been observed experimentally should be sought at such surfaces and interfaces rather than in the bulk of MAPbI$_3$.

In fact, it is worth reconciling our results with recent reports of giant Rashba splitting in MAPbBr$_3$ single crystals using angle-resolved photoelectron spectroscopy (ARPES)[51] and Rashba splitting in CsPbBr$_3$ nanocrystals[52]. ARPES is well known to probe electronic states within 1−10 nm of the surface, where the surface Rashba effect dominates due to the long Debye screening length, as noted above. We thus argue that ref. [51] accurately measures the surface Rashba effect, and the large Rashba splitting observed is not indicative of the bulk bandstructure of the material. Similarly, since the nanocrystals measured in ref. [52] are ~5 nm in diameter, their electronic and optical properties are expected to be dominated by surface Rashba effects.

We additionally note that previous work extracted effective masses through parabolic fits of bandstructures with strong Rashba splitting[26,40]. However, in the presence of a strong Rashba effect transport cannot be described by a free carrier approximation due to the presence of a spin texture[53], and extracting effective masses is an ill-defined procedure due to the non-parabolic dispersions. Since our MA-relaxed structures exhibit a very small Rashba splitting, we can obtain effective masses for a range of MA ion orientations. The electron and hole effective masses of the 10 MA-relaxed tetragonal and cubic structures discussed above are shown in Fig. 4 The computed effective

masses are in excellent agreement with recent experiments[54]. They change appreciably for different MA ion random orientations in spite of the fact that the PbI framework is identical in all the structures, thus justifying the spread in computed effective mass values reported in the literature. While the cubic phase exhibits electron masses lighter than hole masses (by ~50%) as is common in semiconductors, we find an unconventional result in the tetragonal phase, where the electron and hole masses are nearly identical in each of the random MA orientations. These results appear to support a hypothesis advanced recently of balanced transport between the electrons and holes in tetragonal MAPbI$_3$[55,56], though accurate calculations of the mobility are necessary to further ascertain this aspect.

Finally, it is interesting to compare the carrier dynamics in MAPbI$_3$ with perhaps the best-performing conventional solar cell semiconductor, GaAs. The effective masses found here for tetragonal MAPbI$_3$ are comparable with those of electrons and light holes in GaAs[57]—the electron and light hole masses in GaAs are slightly smaller than those in MAPbI$_3$, but the heavy hole masses in GaAs are considerably larger[57]. Commonly reported values for the non-radiative ($\tau_{nr} = 100$ ns) and radiative ($\tau_{rad} = 2$ μs) carrier lifetimes in MAPbI$_3$[48], combined with the effective masses, give estimated diffusion lengths in the range of 1–10 μm, similar to values reported experimentally[13–17] and comparable with those of moderately doped GaAs[58]. Finally, both GaAs and tetragonal MAPbI$_3$ luminesce strongly and with a high quantum yield in surface passivated samples, clearly suggesting that the band gap of MAPbI$_3$ is direct[14]. The parallel between tetragonal MAPbI$_3$ and GaAs, two materials defining a gold standard for solar cells, is intriguing and worthy of further investigation.

In summary, our measurements establish that tetragonal MAPbI$_3$ at room temperature is centrosymmetric with space group $I4/mcm$. Our calculations show that the bulk Rashba effect is very small in MAPbI$_3$ structures with symmetry consistent with experiment, and the band gap nearly direct. Greatly overlooked structural artifacts due to DFT relaxation result in a fictitiously large bulk Rashba effect due to structural distortions of the PbI framework, both in tetragonal and cubic MAPbI$_3$. The DFT fully relaxed structures are inconsistent with the centrosymmetric point groups of these phases, calling for a critical revision of existing bandstructure calculations. While the Rashba effect[26,27] has been invoked in order to explain the favorable transport properties of tetragonal MAPbI$_3$, we show that SOC effects are negligible, and argue that the favorable transport in MAPbI$_3$ is mainly the result of its intrinsic small effective masses and long carrier lifetimes, similar to GaAs.

## Methods

**DFT calculations.** We carry out DFT calculations within the local density approximation (LDA)[59] using the QUANTUM ESPRESSO code[60]. The calculations employ fully relativistic ultrasoft pseudopotentials[61] that include the SOC, a kinetic energy cutoff of 60 Ry for the plane-wave basis set and a uniform $6 \times 6 \times 6$ **k**-point grid. Calculations are performed on both the cubic and tetragonal phases of MAPbI$_3$; both single-formula-unit structures and $\sqrt{2} \times \sqrt{2} \times 2$ supercells with 4 formula units are employed. In the supercells, the initial structures are prepared using experimental lattice parameters, and each of the 4 methylammonium (MA) ions is oriented randomly in one of the 8 equivalent $\langle 111 \rangle$ directions[19]. The structures are relaxed with and without symmetry constraints within the LDA until the force on each atom is less than 0.1 eV/Å. For calculations where only the MA ions are relaxed and the PbI framework is fixed, the lattice parameters are chosen as the ones minimizing the DFT forces for the given symmetry. In these calculations, the PbI framework is fixed and the MA ions are allowed to relax until the average force on each atom less than 0.1 eV/Å. Since LDA plus SOC is known to drastically underestimate the band gap, for bandstructure calculations employed within the LDA an empirical scissor shift is applied in Figs. 3 and 4 to match the experimental band gap. The effective masses of the electrons and holes are extracted using fine grids near the conduction band minimum and valence band maximum, respectively. The DFT calculations using the HSE hybrid exchange-correlation functional are performed using the Vienna Ab initio Simulation Program (VASP) code[62] and projector augmented wave (PAW)[63] pseudopotentials. A kinetic energy cutoff of 400 eV and uniform $6 \times 6 \times 6$ **k**-point grids are employed, together with the SOC, as implemented in the VASP code.

**Crystal growth.** MAPbI$_3$ single crystals were grown by the inverse temperature crystallization method as previously reported[64]. Briefly, 1 M MAPbI$_3$ solution was first prepared by dissolving equimolar MAI and PbI$_2$ in gamma-butyrolactone (GBL) at 70 °C. The temperature of the as-prepared solution was then step-wise increased from 70 to 130 °C with a rate of 5 °C/h to crystallize MAPbI$_3$ by utilizing the inverse solubility of MAPbI$_3$ in GBL. As-grown MAPbI$_3$ crystals were collected from the hot solution with filter papers to remove remaining solution on the surface. The MAPbI$_3$ crystals were further polished using a Unipol1210 polisher with SiC papers (grit size of 600–200) to produce smooth surfaces for the SHG study. The photograph of a sample is given in the Supplementary Fig. 3.

**SHG-RA measurements.** SHG-RA measurements are performed on the polished (100) surface of MAPbI$_3$ at room temperature using a fast rotating scattering plane based technique (see ref. [33]). The incident light is derived from a regeneratively amplified Ti:sapphire laser system operating at a 100 kHz repetition rate with a center wavelength of 800 nm and a pulse duration of 100 fs. The incident fluence is maintained at below 1.5 mJ/cm$^2$ and samples are never heated above the structural phase transition temperature of 327 K to avoid sample damage.

**Expressions for SHG-RA patterns.** The mathematical expressions for the electric-dipole induced SHG-RA patterns for a surface point group $mm2$ are listed below for different polarization geometries:

$$I^{mm2}_{P_{in}-P_{out}}(\phi) = \sin^2(\theta)\left[\chi_{zzz}\sin^2(\theta) + \cos^2(\theta)\left\{(2\chi_{xzx} + \chi_{zxx})\cos^2(\phi) + \left(2\chi_{yyz} + \chi_{zyy}\right)\sin^2(\phi)\right\}\right]^2, \quad (3)$$

$$I^{mm2}_{S_{in}-P_{out}}(\phi) = \sin^2(\theta)\left[\chi_{zyy}\cos^2(\phi) + \chi_{zxx}\sin^2(\phi)\right]^2, \quad (4)$$

$$I^{mm2}_{P_{in}-S_{out}}(\phi) = \frac{1}{4}\sin^2(2\theta)\left[\chi_{xzx} - \chi_{yyz}\right]^2\sin^2(2\phi), \quad (5)$$

$$I^{mm2}_{S_{in}-S_{out}}(\phi) = 0, \quad (6)$$

where $\phi$ and $\theta$ are the scattering plane angle and angle of laser incidence, respectively. Using the experimental value of $\theta = 10°$, a best fit of Eqs. 3, 4, and 5 to the data in Fig. 1 yields the following parameters:

$$\chi_{xzx} = 2.66041 \quad (7)$$

$$\chi_{yyz} = -1.44669 \quad (8)$$

$$\chi_{zyy} = -0.0333659 \quad (9)$$

$$\chi_{zxx} = -4.75148 \quad (10)$$

$$\chi_{zzz} = 161.593 \quad (11)$$

The mathematical expressions for the electric-dipole induced SHG-RA patterns for a bulk point group $4mm$ along the [100] direction are listed below for different polarization geometries:

$$I^{4mm}_{P_{in}-P_{out}}(\phi) = \cos^2(\theta)\cos^2(\phi)\left[(2\chi_{xxz} + \chi_{zxx})\sin^2(\theta) + \cos^2(\theta)\left\{\chi_{zzz}\cos^2(\phi) + (2\chi_{xxz} + \chi_{zxx})\sin^2(\phi)\right\}\right]^2 \quad (12)$$

$$I^{4mm}_{S_{in}-P_{out}}(\phi) = \cos^2(\theta)\cos^2(\phi)\left[\chi_{zxx}\cos^2(\phi) + \left(-2\chi_{xxz} + \chi_{zzz}\right)\sin^2(\phi)\right]^2 \quad (13)$$

$$I^{4mm}_{P_{in}-S_{out}}(\phi) = \sin^2(\phi)\left[\chi_{zxx}\sin^2(\theta) + \cos^2(\theta)\left((-2\chi_{xxz} + \chi_{zzz})\cos^2(\phi) + \chi_{zxx}\sin^2(\phi)\right)\right] \quad (14)$$

$$I^{4mm}_{S_{in}-S_{out}}(\phi) = \sin^2(\phi)\left[(2\chi_{xxz} + \chi_{zxx})\cos^2(\phi) + \chi_{zzz}\sin^2(\phi)\right]^2 \quad (15)$$

**Data availability.** The authors declare that the data supporting the findings of this study are available within the paper and its Supplementary Information files.

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

## Acknowledgements

M.B. acknowledges partial support from start-up funds and from the Space Solar Program Initiative at the California Institute of Technology. K.F. thanks the California Institute of Technology for support through the SURF fellowship program, and Peter Foley and Linn Leppert for fruitful discussions. SHG-RA measurements were supported by the U. S. Department of Energy under grant DE-SC0010533. D.H. also acknowledges funding for instrumentation from the Institute for Quantum Information and Matter, an NSF Physics Frontiers Center (PHY-1125565) with support of the Gordon and Betty Moore Foundation through grant GBMF1250. O.M.B and W.P. acknowledge the support of KAUST. J.B.N. and S.G.L. were supported by the U.S. Department of Energy, Director, Office of Science, Office of Basic Energy Sciences, Materials Sciences and Engineering Division, under Contract No. DE-AC02-05CH11231, through the Theory FWP (KC2301) at Lawrence Berkeley National Laboratory (LBNL). This research used resources of the National Energy Research Scientific Computing Center, a DOE Office of Science User Facility supported by the Office of Science of the U.S. Department of Energy under Contract No. DE-AC02- 05CH11231.

## Author contributions

M.B. conceived the project. K.F. and M.B. carried out the calculations and prepared the manuscript with input from co-authors. W.P. and O.B. synthesized and provided the samples. T.D., J.H., and D.H. designed and conducted the experiments. B.B. contributed to the calculations. J.B.N. and S.G.L. contributed to conceive the theory and calculations. All authors edited the manuscript.

## Additional information

**Competing interests:** The authors declare no competing interests.

