## [Peer Review File · Nature Communications]

Reviewers' comments:

Reviewer #1 (Remarks to the Author):

Report on the manuscript by Kyle Fronha et al "Inversion Symmetry and Bulk Rashba Effect in ..."
The paper by K. Fronha and co-workers reports on experimental and theoretical study of the bulk crystal structure of methylammonium lead iodide perovskite (MAPbI₃). The crystal structure of this interesting photovoltaic material is still a subject of dispute which justify the research. The question of principal importance is whether the bulk crystal structure is centrosymmetric or non-centrosymmetric which implies serious consequences about the electronic structure and Rashba effects in this material.

Experimental part of the paper deals with the measurements of optical second harmonic generation (SHG) reflected from a MAPbI₃ single crystal. This method is widely recognized as being able to distinguish between centrosymmetric and non-centrosymmetric crystal structures with the assumption that SHG is of the electric dipole character. Experimental data shown in Fig. 1 confirm the presence of reliable SHG signals. However, the careful analysis of rotational anisotropies with different combinations of incident (pump light) and outgoing (SHG) polarizations allowed the authors to make a conclusion that the observed SHG signals are related only to the sample surface but not to the inversion symmetry breaking in the bulk crystal. The observed SHG rotational anisotropy pattern is incompatible with the non-centrosymmetric space group I4cm suggested in publications of other authors.

Concerning this part, I have the following comments:

1. It is well known that SHG rotational anisotropy patterns are defined by the point groups. I wonder why the authors discuss in the paper the space groups, p. 5, "Experimental Determination of the Space Group of MAPbI₃" ?
2. SHG experiments were done in reflection geometry, when the inversion symmetry is broken on the light scattering plane. But the conclusion from this observation was done about the bulk crystal symmetry. To my mind, experiments in transmission SHG would provide more reliable conclusion on the bulk symmetry of MAPbI₃ crystal. The authors recognize this by writing, p. 5, "The bulk electric-dipole (ED) induced SHG is only allowed in non-centrosymmetric crystals...". The absence of bulk SHG would be a better prove of centrosymmetric structure (but not final, see 3).
3. As a rule, and very often, the presence or absence of SHG signals is the question of the sample quality, strains, sensitivity of the experimental technique, etc. In fact, some SHG signal can be always detected by using a good technique. But the point is not the detection of SHG, because some signal can be always found. The problem is to find and to prove what is the source of detected SHG signals which may be both intrinsic and/or extrinsic.
4. Only the electric dipole contribution to SHG is analyzed in the paper whereas other contributions may interfere, e.g. the quadrupole contribution.
5. There are publications of other authors which claim the presence of SHG in similar crystals and therefore the question about the true symmetry of MAPbI₃ still remains unsolved.

To my mind, the theoretical part is the main part of the paper. I dare to say that this more extended part, in its own meaning, is independent of the experimental part discussed above. It is devoted to a detailed analysis of the crystal structure and electronic band structure of MAPbI₃ which is based on the DFT calculations with the use of different protocols. The authors realize well that DFT is a method applicable only to zero temperature, but nevertheless it is extensively used as a starting point with subsequent structural relaxations either MA groups or Pb ions. At this point I would like to note that the movement of these groups or Pb ions from the inversion center can only break the local symmetry but it does not mean "a generation of a large ferroelectric distortion" (p. 9). The breaking of bulk inversion does not mean ferroelectricity (example quartz). Calculations allowed the authors to make a conclusion that the most probable crystal structure of MAPbI₃ is centrosymmetric which rules out both the large static and dynamic Rashba effects. Concluding, I must say that the paper deals with a very specific task not of a broad interest. Moreover, I do not feel that all controversies about the true crystal and electronic structure of MAPbI₃ reported in this paper and other literature papers are finally solved. Remaining contradictory results and points of view do not allow me to recommend the paper for publishing in

Nature Communications.

Nevertheless, the results obtained, and in particular the theoretical calculations, are well-grounded, and can be regarded as an alternative view on the symmetry, the crystal structure, and electronic band structure of MAPbI₃ with subsequent consequences related to the Rashba effects. With some modifications concerning the SHG experiments and some other points in the paper it can be submitted to another journal specializing in condensed matter physics.

Reviewer #2 (Remarks to the Author):

I have read the paper titled "Inversion Symmetry and Bulk Rashba Effect in Methylammonium Lead Iodide Perovskite Single Crystals" by Bernardi et al. By means of a combined theoretical/experimental procedure based on DFT/SHG approach, Authors here demonstrate that MAPbI₃ (cubic/tetra/ortho) is centrosymmetric and that accordingly the previously reported dynamic/static Rashba effects (mainly associated with a NON-centrosymmetric space-group) are essentially an artifact of DFT that derives from the impossibility of reproducing symmetry different from the zero Kelvin one (i.e. cubic/tetragonal versus orthorhombic).

The results are novel and the approach is pretty convincing and accordingly I would recommend the publication of the paper.

There remain anyway some points that Authors should have to better clarify before the paper can be published.

-It is not clear to me from the text if authors focused also on the orthorhombic polymorph. And this is of relevance I think. I mean, to better assess their conclusions, I feel authors should better focus on the such phase for which (it is the zero K structure) no thermal effects are present by definition. According to the explanation provided by the Authors, I would indeed expect that a full DFT-based optimization of the orthorhombic cell (no need of any constraints) should lead to a symmetry which is centrosymmetric (negligible Rashba effects) and also in agreement with other experimental data: this result would strengthen the results already obtained for cubic and tetragonal phases.

-Recently, I have bumped into some stressed reviewers who asked for exotic different computational flavors, far from my proposed, even if they clearly know that results are never completely changed by the different flavor and aware of the fact that trends are mostly kept. In this sense, I would not ask for further different calculations. At the same time, I wonder if authors can comment the choice of not including (as far as I get from the "Methods" section) any nonlocal vdW functional which, to the best of my knowledge, is quite impactful in the treatment of such systems.

-I am not a big fan of partial optimizations. But I understand that in the present analysis partial optimization becomes almost mandatory. Anyway, keeping some coordinates as they are experimentally reported while optimizing others in the same system may give rise to some stress which in my opinion can play some (minor?major?) impact on the bandplot profile. Can Authors comment on this aspect?

Minor points:

-Provide .cif files for the centrosymmetric structures of relevance.

-I would recommend the Authors to add a Table along with Figure 1 in order to better describe bulk/surface point groups of centrosym/non-centrosym groups. It would be more straightforward and easy to get for the reader.

-On pag 2, immediately after the Abstract section. Once the acronym is decided and introduced (MAPbI) it should be kept along the whole paper. It is confusing and stylistically not very nice

immediately after the acronym definition to use once more MAPbI₃ and immediately after once more MAPbI and so on... What I would do is to define at first methylammonium lead iodide as "MAPbI₃" and in parentheses then add "hereafter also MAPbI" or whatever the Authors prefer.

Reviewer #3 (Remarks to the Author):

This paper describes a multi-pronged investigation into the nature of inversion symmetry in MAPbI₃ and its impact on whether or not there can be a strong bulk Rashba effect. The authors conclude that the perovskite has space group I4/mcm, and this effectively does not allow it to have a strong Rashba effect. They show how other groups may have misinterpreted DFT relaxation to imply a strong Rashba effect, but that this is inconsistent with crystal structure. The paper appears to be of high scientific merit and has been carried out carefully and thoroughly. It is well written and easy to follow. The results are novel and important to add to the ongoing discussion regarding some of the apparently anomalous properties of this perovskite in the field. I therefore recommend publication once the following minor issue has been dealt with:

1. Other groups have seen experimental evidence for large Rashba splitting in the halide perovskites, both at low temperature and room temperature. This work appears to have overlooked some of the more-cited literature pertaining to this. To avoid confusing the field further, the authors should discuss how their work relates to in particular the two papers listed below, and suggest why their results do not agree with the conclusions of these papers:

- <https://journals.aps.org/prl/abstract/10.1103/PhysRevLett.117.126401>
- <http://pubs.acs.org/doi/pdf/10.1021/acs.nanolett.7b02248>

REVIEWER REPORTS

REFEREE 1

Introductory Comment:

Report on the manuscript by Kyle Fronha et al “Inversion Symmetry and Bulk Rashba Effect in ...” The paper by K. Fronha and co-workers reports on experimental and theoretical study of the bulk crystal structure of methylammonium lead iodide perovskite (MAPbI₃). The crystal structure of this interesting photovoltaic material is still a subject of dispute which justify the research. The question of principal importance is whether the bulk crystal structure is centrosymmetric or non-centrosymmetric which implies serious consequences about the electronic structure and Rashba effects in this material.

Experimental part of the paper deals with the measurements of optical second harmonic generation (SHG) reflected from a MAPbI₃ single crystal. This method is widely recognized as being able to distinguish between centrosymmetric and non-centrosymmetric crystal structures with the assumption that SHG is of the electric dipole character. Experimental data shown in Fig. 1 confirm the presence of reliable SHG signals. However, the careful analysis of rotational anisotropies with different combinations of incident (pump light) and outgoing (SHG) polarizations allowed the authors to make a conclusion that the observed SHG signals are related only to the sample surface but not to the inversion symmetry breaking in the bulk crystal. The observed SHG rotational anisotropy pattern is incompatible with the non-centrosymmetric space group I4cm suggested in publications of other authors.

Response to introductory comment:

We thank the Referee for the feedback and comments. We have fully taken into account the referee’s remarks, and address the technical concerns regarding the SHG experiments below.

Comment #1:

It is well known that SHG rotational anisotropy patterns are defined by the point groups. I wonder why the authors discuss in the paper the space groups, p. 5, “Experimental Determination of the Space Group of MAPbI₃”?

Response to comment #1:

The Referee is correct that SHG is only sensitive to the point group symmetry. We meant to write “Determination of the Point Group” in fact, and this was really a slip of the pen.

Changes made in response to comment #1:

On page 5, we have changed the text to “Determination of the Point Group of MAPbI₃”.

Comment #2:

SHG experiments were done in reflection geometry, when the inversion symmetry is broken on the light scattering plane. But the conclusion from this observation was done about the bulk crystal symmetry. To my mind, experiments in transmission SHG would provide more reliable conclusion on the bulk symmetry of MAPbI₃ crystal. The authors recognize this by writing, p. 5, “The bulk electric-dipole (ED) induced SHG is only allowed in non-centrosymmetric crystals...”. The absence of bulk SHG would be a better prove of centrosymmetric structure (but not final, see 3).

Response to comment #2:

Since 1.5 eV light has a long (several micron) penetration depth into MAPbI₃ [see Zhou et al., Nature Communications 7, 11193 (2016)], our SHG experiments are sensitive to both the surface and bulk. Reflection based SHG experiments are commonly used to measure the bulk

symmetry of crystals [see for example Harter et al., Science 356, 295 (2017)]. Transmission experiments would need to be done on an optically thin sample of a few microns, which is currently not feasible.

Comment #3:

As a rule, and very often, the presence or absence of SHG signals is the question of the sample quality, strains, sensitivity of the experimental technique, etc. In fact, some SHG signal can be always detected by using a good technique. But the point is not the detection of SHG, because some signal can be always found. The problem is to find and to prove what is the source of detected SHG signals which may be both intrinsic and/or extrinsic.

Response to comment #3:

We completely agree with the Referee that the point is not the detection of SHG but rather the identification of the source of the SHG. This is precisely why we measured the RA curves: to determine the source of the SHG that we detected. We concluded that the source was consistent with surface electric-dipole (ED) and not consistent with bulk ED.

Comment #4:

Only the electric dipole contribution to SHG is analyzed in the paper whereas other contributions may interfere, e.g. the quadrupole contribution.

Response to comment #4:

The fact that the measured SHG-RA patterns do not fit the patterns expected from a bulk electric-dipole (ED) contribution is sufficient to make a determination about bulk inversion symmetry. While a bulk electric-quadrupole (EQ) contribution is in principle allowed even from a centrosymmetric crystal, it is in general greatly suppressed with respect to the bulk ED contribution by a factor of a/λ , where λ is the optical wavelength and a is the lattice spacing. Moreover, the bulk ED and bulk EQ contributions are governed by completely different susceptibility tensors. Therefore they cannot cancel each other out.

That being said, we did calculate the bulk EQ contribution from the centrosymmetric point group $4/mmm$, and found that a finite response is expected in the S_{in} - S_{out} channel. Since we do not measure any signal in the S_{in} - S_{out} channel (in agreement with surface ED), we can confidently rule out a bulk EQ contribution.

Comment #5:

There are publications of other authors which claim the presence of SHG in similar crystals and therefore the question about the true symmetry of MAPbI₃ still remains unsolved.

Response to comment #5:

The presence of SHG does not itself determine inversion symmetry (see the Referee's own point in comment #3 above). As we have shown, surface SHG is always present. One must be able to disentangle surface from bulk SHG in order to determine if bulk inversion symmetry is absent, and this is precisely what we are reporting here.

Comment #6:

To my mind, the theoretical part is the main part of the paper. I dare to say that this more extended part, in its own meaning, is independent of the experimental part discussed above. It is devoted to a detailed analysis of the crystal structure and electronic band structure of MAPbI₃ which is based on the DFT calculations with the use of different protocols. The authors realize well that DFT is a method applicable only to zero temperature, but nevertheless it is extensively used as a starting point with subsequent structural relaxations

either MA groups or Pb ions. At this point I would like to note that the movement of these groups or Pb ions from the inversion center can only break the local symmetry but it does not mean “a generation of a large ferroelectric distortion” (p. 9). The breaking of bulk inversion does not mean ferroelectricity (example quartz). Calculations allowed the authors to make a conclusion that the most probable crystal structure of MAPbI₃ is centrosymmetric which rules out both the large static and dynamic Rashba effects.

Response to comment #6:

We are pleased that the referee finds merit in our theoretical and computational results. We agree that the breaking of bulk inversion symmetry does not imply ferroelectricity, and have revised the manuscript accordingly.

Changes made in response to comment #6: On page 11 of the revised manuscript, we have changed the sentence mentioned by the Referee (which was previously on page 9) to “generates a large structural distortion.”

Concluding Comment:

Concluding, I must say that the paper deals with a very specific task not of a broad interest. Moreover, I do not feel that all controversies about the true crystal and electronic structure of MAPbI₃ reported in this paper and other literature papers are finally solved. Remaining contradictory results and points of view do not allow me to recommend the paper for publishing in Nature Communications.

Nevertheless, the results obtained, and in particular the theoretical calculations, are well-grounded, and can be regarded as an alternative view on the symmetry, the crystal structure, and electronic band structure of MAPbI₃ with subsequent consequences related to the Rashba effects. With some modifications concerning the SHG experiments and some other points in the paper it can be submitted to another journal specializing in condensed matter physics.

Response to concluding comment:

We truly value the Referee’s feedback, which has helped us improve the manuscript. We maintain that our work is a significant step forward toward solving the controversies on the structure and Rashba effect in MAPbI₃. We have addressed the technical concerns regarding the SHG measurements above. The combination of the measurements and the theoretical work, in which the Referee finds merit, are highly relevant to the metal-halide perovskite community. In particular, we establish that the Rashba splitting is negligible in the bulk, and that the band gap is nearly direct, which is a point of major debate in the community. We also establish the correct structures of the tetragonal and cubic phases to use as a single snapshot in first-principles calculations, an essential result to compute from first principles charge transport and optical properties. As stated in the journal scope, Nature Communications publishes work that “represents important advances of significance to specialists within each field.” On this basis, we believe that our work is a good match for the journal.

REFeree 2

Introductory Comment:

I have read the paper titled "Inversion Symmetry and Bulk Rashba Effect in Methylammonium Lead Iodide Perovskite Single Crystals" by Bernardi et al. By means of a combined theoretical/experimental procedure based on DFT/SHG approach, Authors here demonstrate that MAPbI₃ (cubic/tetra/ortho) is centrosymmetric and that accordingly the

previously reported dynamic/static Rashba effects (mainly associated with a NON-centrosymmetric space-group) are essentially an artifact of DFT that derives from the impossibility of reproducing symmetry different from the zero Kelvin one (i.e. cubic/tetragonal versus orthorhombic).

The results are novel and the approach is pretty convincing and accordingly I would recommend the publication of the paper.

There remain anyway some points that Authors should have to better clarify before the paper can be published.

Response to introductory comment:

We thank the Referee for the positive feedback on our work and for recommending publication in Nature Communications after minor revisions. We are glad that the Referee is convinced by our methodology and sees merit in our results. We have revised the manuscript to fully address the Referee's comments and suggestions.

Comment #1:

It is not clear to me from the text if authors focused also on the orthorhombic polymorph. And this is of relevance I think. I mean, to better assess their conclusions, I feel authors should better focus on the such phase for which (it is the zero K structure) no thermal effects are present by definition. According to the explanation provided by the Authors, I would indeed expect that a full DFT-based optimization of the orthorhombic cell (no need of any constraints) should lead to a symmetry which is centrosymmetric (negligible Rashba effects) and also in agreement with other experimental data: this result would strengthen the results already obtained for cubic and tetragonal phases.

Response to comment #1:

We thank the Referee for the insight regarding the orthorhombic phase of MAPbI₃. It is a sensible suggestion and we have fully taken it into account.

Since the tetragonal and cubic phases are those relevant for most applications in solar cells and optoelectronics, the vast majority of calculations on MAPbI₃ are carried out on structures that are cubic or tetragonal before relaxation. Yet, the referee is correct that should the starting structure be the experimentally determined low temperature orthorhombic phase, upon relaxation, one should reach a structure that is still centrosymmetric. This is indeed the case as has been reported in the literature (see Filip et al. *J. Phys. Chem. C*, 2015, 119, pp 25209–2521). The authors use ultrasoft, fully relativistic pseudopotentials and LDA when relaxing and calculating the bandstructure, as we do in our work. They report no discernible Rashba splitting in their bandstructures, suggesting a centrosymmetric structure.

To verify this result, we have calculated the bandstructure of the orthorhombic phase. Starting from a structure measured experimentally, we carried out a full relaxation of the unit cell and atomic positions, without constraints. We find that indeed, as the Referee suggests, the structure is still centrosymmetric at the end of the relaxation, and no Rashba splitting is observed. We mention this result in the revised manuscript, and include the calculated bandstructure in the Supporting Information.

Changes made in response to comment #1:

We have included a sentence on page 11 “Note that if the low temperature orthorhombic structure is fully relaxed, the octahedra remain undistorted and no Rashba splitting is observed.⁴² We verify this result by fully relaxing an experimentally measured orthorhombic structure,⁴³ and find that the structure remains centrosymmetric upon relaxation. Its bandstructure, which is given in Supporting Fig. S2, does not exhibit a Rashba splitting at the band edges.”

Here, Ref. 42 is the paper by Filip et al. mentioned above and Ref. 43 a paper by Whitfield et al., *Scientific Reports*, 6, 35685 (2016), who measured the orthorhombic structure at low temperature. We included the bandstructure in Fig. S2 of the Supporting Information.

Comment #2:

Recently, I have bumped into some stressed reviewers who asked for exotic different computational flavors, far from my proposed, even if they clearly know that results are never completely changed by the different flavor and aware of the fact that trends are mostly kept. In this sense, I would not ask for further different calculations. At the same time, I wonder if authors can comment the choice of not including (as far as I get from the "Methods" section) any nonlocal vdW functional which, to the best of my knowledge, is quite impactful in the treatment of such systems.

Response to comment #2:

We agree with the Referee that including a vdW functional can be important in halide perovskites. Since the main point here is the ability of DFT to capture the finite temperature phases, which are stabilized by the vibrational entropy of the PbI framework, our results are not affected by including the vdW interactions in the relaxation. The vdW functional – whether semiempirical Grimme, fully nonlocal, or Tkatchenko-Scheffler – does not contain the physics needed to stabilize the high-temperature phases and prevent their transition toward the orthorhombic phase upon relaxation. In any case, we thank the Referee for making this point, and agree that future work should use vdW functionals for added accuracy.

Comment #3:

I am not a big fan of partial optimizations. But I understand that in the present analysis partial optimization becomes almost mandatory. Anyway, keeping some coordinates as they are experimentally reported while optimizing others in the same system may give rise to some stress which in my opinion can play some (minor?major?) impact on the bandplot profile. Can Authors comment on this aspect?

Response to comment #3:

We thank the Referee for this comment about the potential drawbacks of partial optimizations, which could result in residual stress affecting the computed bandstructures. We are also glad that the Referee understands that in our analysis such partial geometry optimizations are needed to adopt the experimental crystal symmetry.

In our calculations, we minimize the residual stress by first relaxing the unit cell and atomic positions without constraints. As shown in Fig. 2, in the first 5–10 steps the structure retains the initial crystal symmetry, but greatly minimizes the stress; in particular, since ours are LDA calculations, the lattice constants decrease, but at first maintain their initial ratios. If the structure starts deviating from the initial symmetry (and the lattice constants changing their mutual ratios), say, after N relaxation steps, we fix the lattice constants to the values at step $N-1$, which ensures that the stress is minimized. Starting from these “low pressure” structures (this is in fact how we call them), we fix the PbI framework and relax the MA ions. The resulting structures are employed to calculate the “MA relaxed” bandstructures in Figure 3. This detail is briefly discussed in the Methods section, where we state: “the lattice parameters are chosen as the ones minimizing the DFT forces for the given symmetry”.

Comment #4:

Provide .cif files for the centrosymmetric structures of relevance.

Changes made in response to comment #4:

We thank the referee for this suggestion, which helps improve the transparency and reproducibility of our work. We have provided .cif files for the centrosymmetric cubic and tetragonal phases of MAPbI₃ along with the resubmitted manuscript. A .cif file with the fully relaxed orthorhombic structure, as suggested by the Referee, is also included.

Comment #5:

I would recommend the Authors to add a Table along with Figure 1 in order to better describe bulk/surface point groups of centrosym/non-centrosym groups. It would be more straightforward and easy to get for the reader.

Response to comment #5:

We agree that a table highlighting bulk and surface point groups for both the centrosymmetric *I4/mcm* and non-centrosymmetric *I4cm* space groups would make the data in Figure 1 more easily understood. We have added such a table in the revised manuscript (see Table 1 on page 7).

Changes made in response to comment #5:

A table containing the bulk and surface point groups for the *I4/mcm* and *I4cm* tetragonal MAPbI₃ space groups is included as Table 1 on page 7 of the revised manuscript. A reference to this table is made on page 6: "The space groups and associated point groups are summarized in Table 1."

Comment #6:

On pag 2, immediately after the Abstract section. Once the acronym is decided and introduced (MAPbI) it should be kept along the whole paper. It is confusing and stylistically not very nice immediately after the acronym definition to use once more MAPbI3 and immediately after once more MAPbI and so on... What I would do is to define at first methylammonium lead iodide as "MAPbI3" and in parentheses then add "hereafter also MAPbI" or whatever the Authors prefer.

Response to comment #6:

We thank the referee for noticing this unintentional switching of acronyms, and have addressed this issue in the revised manuscript. We are truly grateful for the feedback from the Referee, which has greatly helped us improve the manuscript.

Changes made in response to comment #6:

We have changed all of the acronyms referring to methylammonium lead iodide in the manuscript to MAPbI₃, removing the alternative acronym MAPbI to eliminate any confusion.

REFEREE 3

Introductory Comment:

*This paper describes a multi-pronged investigation into the nature of inversion symmetry in MAPbI₃ and its impact on whether or not there can be a strong bulk Rashba effect. The authors conclude that the perovskite has space group *I4/mcm*, and this effectively does not allow it to have a strong Rashba effect. They show how other groups may have misinterpreted DFT relaxation to imply a strong Rashba effect, but that this is inconsistent*

with crystal structure. The paper appears to be of high scientific merit and has been carried out carefully and thoroughly. It is well written and easy to follow. The results are novel and important to add to the ongoing discussion regarding some of the apparently anomalous properties of this perovskite in the field. I therefore recommend publication once the following minor issue has been dealt with:

Response to introductory comment:

We are grateful for the very positive comments and for recommending publication in Nature Communications. We address below and in the revised manuscript the Referee's comment.

Comment #1:

Other groups have seen experimental evidence for large Rashba splitting in the halide perovskites, both at low temperature and room temperature. This work appears to have overlooked some of the more-cited literature pertaining to this. To avoid confusing the field further, the authors should discuss how their work relates to in particular the two papers listed below, and suggest why their results do not agree with the conclusions of these papers:

- <https://journals.aps.org/prl/abstract/10.1103/PhysRevLett.117.126401>

- <http://pubs.acs.org/doi/pdf/10.1021/acs.nanolett.7b02248>

Response to comment #1:

We thank the Referee for drawing our attention to these two works on experimental measurement of the Rashba splitting in halide perovskites.

The first paper, by Niesner et al., finds a giant Rashba splitting in MAPbBr₃ at low temperature using angle resolved photoemission spectroscopy (ARPES). ARPES is well known to probe electronic states within 1–10 nm of the surface. We know that inversion symmetry is necessarily broken at the surface, and that the surface Rashba effect is only weakly screened given the long Debye screening length (up to 1–100 μm, see page 17 of our manuscript), which is much longer than the depth probed using ARPES. We conclude that the ARPES measurements probe the surface Rashba effect, and the large Rashba splitting observed is not indicative of the bulk bandstructure of the material.

The second work, by Isarov et al., focuses on the Rashba effect in colloidal perovskite nanocrystals. Since the nanocrystals diameter is of order ~5 nm, surface Rashba effects are expected to dominate the optical properties measured in the paper. The authors of this work indeed mention that the surface of the colloidal nanocrystals is responsible for the Rashba effect; we similarly conclude that such experiments on nm-size nanocrystals effectively probe the surface Rashba effect rather than the bulk electronic structure.

It is a valuable suggestion that not mentioning these works explicitly could prove confusing to the field. We cite and briefly discuss these two papers in our revised manuscript.

Changes made in response to comment #1:

We have included a paragraph on page 17, where we cite both papers mentioned by the Referee and briefly explain their relation to our results. The new paragraph reads:

“In fact, it is worth reconciling our results with recent reports of giant Rashba splitting in MAPbBr₃ single crystals using angle-resolved photoelectron spectroscopy (ARPES)⁵² and Rashba splitting in CsPbBr₃ nanocrystals.⁵³ ARPES is well known to probe electronic states within 1–10 nm of the surface, where the surface Rashba effect dominates due to the long Debye screening length, as noted above. We thus argue that Ref. 52 accurately measures the surface Rashba effect, and the large Rashba splitting observed is not indicative of the bulk bandstructure of the material. Similarly, since the nanocrystals measured in Ref. 53 are ~5 nm in diameter, their electronic and optical properties are expected to be dominated by surface Rashba effects.”

Here, Ref. 52 is the ARPES paper by Niesner et al. and Ref. 53 the quantum dot paper by Isarov et al.

REVIEWERS' COMMENTS:

Reviewer #1 (Remarks to the Author):

Report of Reviewer #1 on the revised manuscript NCOMMS-17-3172 by Kyle Frohna et al
"Inversion Symmetry ..."

My comments in the previous Report were mostly related to the experimental measurements and interpretation of results on optical second harmonic generation in single crystals of methylammonium lead iodide perovskite. These data and conclusions served as a basis for defining the true crystal structure of this material and for supporting results of the DFT calculations. I did not discuss the theoretical part which was analyzed in detail by two other Reviewers.

The authors recognized the importance and validity of all my comments and questions and presented very detailed and well-argued responses to all of them.

Now I can join recommendations of the two other Reviewers that the manuscript can be accepted for publication in Nature Communication as soon as their detailed comments and questions also get convincing replies and explanations.

Reviewer #2 (Remarks to the Author):

Authors reply is satisfactory and now the paper is extremely improved. Accordingly I can recommend its publication in Nat Commun.

Best wishes.

Reviewer #3 (Remarks to the Author):

The authors have adequately addressed the comments that I made, and in my opinion, they have satisfactorily addressed the points brought up by the other reviewers. I therefore think the work is suitable for publication in Nat Comms.